# Continual Learning After Model Deployment

## Abstract

This paper studies continual learning after model deployment. A real-world application environment is often an open world filled with novel or out-of-distribution (OOD) objects that have not been seen before. We can call continual learning in such an environment *open-world continual learning* (OWCL). OWCL incrementally performs two main tasks: (1) detecting OOD objects, and (2) continually learning the OOD or new objects on the fly. Although OOD detection and continual learning have been extensively studied separately, their combination for OWCL has barely been attempted. This is perhaps because in addition to the existing challenges of OOD detection and continual learning such as *catastrophic forgetting* (CF), OWCL also faces the challenge of data scarcity. As novel objects appear sporadically, when an object from a new/novel class is detected it, is difficult to learn it from one or a few samples to give good accuracy. This paper proposes a novel method called OpenLD to deal with these problems based on *linear discriminant analysis* (LDA) and a pre-trained model. This method enables OOD detection and incremental learning of the detected samples on the fly with no CF. Experimental evaluation demonstrates the effectiveness of OpenLD.

## 1 Introduction

Traditional machine learning operates in a closed world, where everything encountered during testing has already been seen during training, meaning nothing novel can appear after model deployment.(Bendale & Boult, 2015; Liu et al., 2023a). In contrast, the real world is an open environment, full of unknowns and novelties, also referred to as out-of-distribution (OOD) objects. To function effectively in this open world, an AI agent must continuously learn on the fly after deployment, rather than relying on periodic offline retraining initiated by human engineers. We call this paradigm the *open-world continual learning* (**OWCL**). OWCL must have three key capabilities: (1) detecting OOD objects, (2) obtaining the class labels of the detected OOD objects, and (3) learning the OOD objects on the fly incrementally. (3) is the *class-incremental learning* setting of continual learning as the system learns more and more classes. This paper focuses on (1) and (3). As humans, when we see some unknown objects, we usually ask others for their names (i.e., *class labels*). Thus, for (2), we assume the agent can ask humans or other agents to provide labels for the detected OOD objects, but we want to minimize the number of asks. Furthermore, in OWCL, the data comes in a stream and OOD detection and learning of the detected novel objects have to be done online.

**OWCL Problem Setting**. Since OWCL aims to learn continually after model deployment, the initial model $M$ to be deployed is assumed to be well-trained with a set of initial classes $C$ of labeled data. After $M$ is deployed in its application, it detects and learns more and more new classes. At the steady state, the set of all classes that the system has encountered is $C^A = C \cup C^L \cup C^E$, where $C^L$ is the set of new classes that have appeared after model deployment and are *well learned* after seeing a good number of training samples, and $C^E$ is a set of emerging new classes that have been seen but are *not well learned yet*, i.e., not enough labeled training data have been seen to well-learn the classes. We denote $C^+ = C \cup C^L$ as the set of well-learned classes so far and $C^N = C^L \cup C^E$ as the set of all new classes seen after deploying $M$. With incremental learning, $M$ becomes $M^+$, covering all classes in $C^A$. Each iteration of OWCL performs two main functions.

**(1)**. *OOD Detection and Classification*. $M^+$ detects whether an incoming test sample $x$ is OOD. If not, it is classified to one of the classes in $C^+$. OOD classes include those emerging classes in $C^E$ as they still need some more data to be well-learned, but OOD detection can leverage the

Figure 1: Pipeline of the proposed OpenLD method. ID classes (in-distribution classes), which are used in OOD detection, include both $C$ and $C^L$.

already-seen samples of these classes. This is different from existing OOD detection or continual OOD detection (Rios et al., 2022) (which does OOD detection in the process of continual learning).

**(2).** *Incremental Learning*. The system learns each detected OOD/novel sample $x$ after obtaining its class label by asking a human user or another knowledgeable agent. If $x$ assigned a class label in $C$, do nothing (no learning). If $x$ is assigned a class label in $C^N$, the current model $M^+$ is updated.

OWCL is thus related to two main areas of research, OOD detection (Yang et al., 2021b; Chandola et al., 2009) and the class incremental learning (CIL) setting of continual learning (Van de Ven & Tolias, 2019; De Lange et al., 2021; Wang et al., 2023; Mai et al., 2022). These topics have been studied separately and extensively. However, in OWCL, OOD detection has to be done continually unlike the traditional static OOD detection with a set of fixed *in-distribution* (ID) classes. The number of ID classes increases as the agent learns new classes of objects. CIL, which learns a sequence of tasks (where each task consists of a set of classes) incrementally, still faces a major challenge of *catastrophic forgetting* (CF). CF refers to the phenomenon that in learning a new task, the learner needs to modify the parameters learned for previous tasks, which may cause performance degradation for previous tasks. Furthermore, existing CIL methods do not do OOD detection.

A novel approach called OpenLD (*Open*-world continual learning via *L*inear discriminant analysis and OOD *D*etection) is also proposed to learn in the OWCL setting, i.e., performing the above two main functions. The method is based on *linear discriminant analysis* (LDA) (Pang et al., 2005). LDA assumes that given the class, the data follows a normal distribution with a mean and covariance matrix. It further makes the assumption that the class covariances are identical, i.e., all classes share one covariance matrix but have different means. LDA uses the means and covariance for classification. However, LDA is not suitable for OOD detection because LDA is based on likelihood ratio, which is only suitable for closed-world classification, as OOD detection needs a measure of absolute distance from a sample to a distribution. In this work, we use Mahalanobis distance (MD) and related methods for OOD detection with a novel idea (see below).

After obtaining the label of each detected OOD sample, OpenLD learns it immediately. Each new class still uses the covariance matrix learned initially in $M$, but the mean of the class is updated. The pipeline of the proposed method is shown in Figure 1. This paper makes the following contributions.

**1.** It proposes a realistic open-world continual learning setting, which is important as an AI agent working in the real open world needs the capability to continually learn new knowledge on the fly after deployment to make it more and more knowledgeable and ultimately achieve learning autonomy, i.e., incremental learning for self-improvement without the involvement of human engineers.

**2.** It proposes a novel approach based on incremental updating of the model with a shared covariance and different means for different classes, which has **no CF** and gives good accuracy without a large number of training samples from each new class. This is particularly important because it is hard to obtain many labeled samples after model deployment.

**3.** In continual OOD detection, OpenLD not only uses the ID classes but also already detected OOD samples to help detect OOD data more accurately. To our knowledge, this has not been done before.

Experiments have been conducted to demonstrate the effectiveness of the proposed OpenLD. The **code of OpenLD** has been submitted in *Supplementary Materials*.

## 2 RELATED WORK

*OOD Detection.* OOD detection has been studied under many names, e.g., novelty detection, outlier or anomaly detection, and open set recognition (Ghassemi & Fazl-Ersi, 2022; Cui & Wang, 2022; Shen et al., 2021; Cortes et al., 2016; Geifman & El-Yaniv, 2017; Andrews et al., 2016; Liu et al., 2008; Tax, 2002; Scheirer et al., 2012; Bendale & Boult, 2015; 2016; Ge et al., 2017; Neal et al., 2018; Malinin & Gales, 2018; 2019; Malinin et al., 2019; Zhou et al., 2023). In recent years, neural network-based approaches have produced state-of-the-art algorithms. They all generate some kinds of OOD scores (Yang et al., 2021b). One popular category of methods uses logits to compute such scores (Hendrycks & Gimpel, 2017; Hendrycks et al., 2019; Wei et al., 2022). Some other works also use additional mechanisms (Liang et al., 2017; Sun et al., 2021; Liu et al., 2020a; Wang et al., 2021; Liu et al., 2023b). Many also improve the architecture and features (DeVries & Taylor, 2018; Yu & Aizawa, 2019; Hsu et al., 2020; Huang & Li, 2021; Shama Sastry & Oore, 2019; Sun & Li, 2022). Yet, some others use ensembles (Lakshminarayanan et al., 2017; Gal & Ghahramani, 2016; Guo et al., 2017). Some approaches also expose the system with some OOD data during training (Hendrycks et al., 2018; Yang et al., 2021a; Chen et al., 2021; Papadopoulos et al., 2021). Some work has been done to cluster the detected OOD samples into potential classes (Han et al., 2021). However, we do not do that as we learn each OOD sample immediately after it is detected.

Our OOD detection method is most related to distance-based methods like Mahalanobis distance (MD) (Lee et al., 2018). Ren et al. (2021) addressed some limitations of MD and proposed the Relative Mahalanobis distance (RMD) method. However, our work does both OOD detection and continual learning. Unlike existing OOD detection methods, in our case, the number of ID classes is not fixed but keeps increasing. We also use newly identified OOD data to detect more OOD data.

*Continual learning.* The existing work mainly focuses on overcoming CF (Castro et al., 2018; Fernando et al., 2017; Kemker & Kanan, 2017). Current methods belong to several categories. *Regularization*-based methods try to deal with CF in learning a new task by using a regularization term to penalize changes to existing network parameters that are important to previous tasks (Kirkpatrick et al., 2017; Zenke et al., 2017; Li & Hoiem, 2016; Ritter et al., 2018; Schwarz et al., 2018; Xu & Zhu, 2018; Lee et al., 2019; Ahn et al., 2019). *Replay*-based methods overcome CF by storing some data from previous tasks in a memory buffer. When learning a new task, the saved data and the new task data are used jointly to train the new task while also adjusting the previous task parameters so that their performance will not deteriorate significantly (Rusu et al., 2016; Lopez-Paz & Ranzato, 2017; Rebuffi et al., 2017; Aljundi et al., 2019a; Chaudhry et al., 2019; Wu et al., 2019; Rolnick et al., 2019; Liu et al., 2020b; Yan et al., 2021; Wang et al., 2022; Guo et al., 2022; Kim et al., 2022a). *Pseudo-replay*-based methods build a data generator that can generate previous task data to be used in learning a new task (Shin et al., 2017; Wu et al., 2018; Seff et al., 2017; Kemker & Kanan, 2018; Ostapenko et al., 2019; Zhu et al., 2021). The generated data replaces the replay data in the replay-based approach. *Parameter-isolation*-based methods learn and use masks to protect the learned models for previous tasks so that they will not be updated in learning a new task, which avoids CF (Mallya & Lazebnik, 2017; Abati et al., 2020; Von Oswald et al., 2019; Rajasegaran et al., 2020; Henning et al., 2021; Serra et al., 2018; Wortsman et al., 2020). *Orthogonal projection*-based methods learn each task in an orthogonal space to the previous task spaces to reduce task interference or CF (Zeng et al., 2019; Chaudhry et al., 2020; Lin et al., 2022). Recently, the parameter-isolation approach and OOD detection are combined for class-incremental learning (CIL) (Kim et al., 2022b). However, this approach is not for open-world continual learning, but for traditional CIL.

Most of the above methods were proposed for *batch continual learning*, meaning that when a task arrives, all its training data are available and the learning can be done in any number of epochs. However, our work is more related to *online continual learning*, which learns from a data stream (Mai et al., 2022). Almost all approaches in online continual learning are based on replay (Chaudhry et al., 2020; Aljundi et al., 2019a; Shim et al., 2021; Prabhu et al., 2020; Aljundi et al., 2019b; Buzzega et al., 2020; Mai et al., 2021; Cha et al., 2021; Pham et al., 2021; Bang et al., 2021; 2022; Koh et al., 2022). Since our method learns after model deployment, we use online continual learning. However, there are some major differences. First, we do not use any replay data. Second, we perform OOD detection and incrementally learn the detected OOD classes on the fly. Our work is most closely related to the work in Hayes & Kanan (2020), which used *incremental LDA* (Pang et al., 2005) for online continual learning, but it does not detect OOD instances or work in open-world continual learning after model deployment. Further, our work does not use incremental LDA.

Some attempts have been made to solve OWCL. Gummadi et al. (2022) proposed the SHELS method for OOD detection and continual learning. However, their method does not truly integrate OOD detection and continual learning like ours. Their two functions can only be evaluated separately and its continual learning is not on streaming data but on traditional batch-based data and training. Bendale & Boult (2015) detected OOD data using SVM and computing the mean of each class for classification. However, for OOD detection, it builds a separate model for each class, which is not in the spirit of continual learning, learning all tasks or classes incrementally in a single model. Rios et al. (2022) proposed the deep learning method (IncDFM) that does OOD detection using a pre-trained model. However, it does not continually learn or integrate with continual learning as we do. Roy et al. (2022) investigated class-incremental novel class discovery (class-iNCD), focusing on strategies for discovering and adapting to new classes over time by training a joint classifier for both base and novel classes. He & Zhu (2022) addressed OOD detection challenges in the unsupervised continual learning setting. None of these methods truly integrated OOD detection and continual learning such that the system can learn in the open world after model deployment.

## 3   PROPOSED APPROACH: OPENLD

The proposed new OWCL setting has been discussed in the introduction section. From this subsection, we present the details. We start with the key ***challenges*** of OWCL and the main idea of the proposed techniques and their ***novelties*** for dealing with the challenges. Recall that OWCL aims to learn after model deployment and the two main steps are: (1) continual OOD detection, and (2) class incremental learning (CIL). As discussed earlier, we assume that the class label for each detected OOD sample can be obtained by asking a human user or another agent while working with them.

Both steps are highly challenging. For (1), continual OOD detection is dynamic in nature. Unlike a traditional OOD detection model, which is built based on a set of fixed ID classes. The key novelty of our OOD detection is that we also use the identified OOD samples to detect more OOD samples. For (2), the key challenge is that the AI agent should not ask human users for labels of the detected OOD samples too many times, which means that the proposed method OpenLD must have a strong learning capability without using many labeled samples. For both (1) and (2), there is also the challenge of *catastrophic forgetting* (CF) in continual learning. OpenLD deals with all these challenges with the help of *linear discriminant analysis* (LDA), which we introduce next.

### 3.1   LINEAR DISCRIMINANT ANALYSIS (LDA)

The original LDA classifies two classes $\{0, 1\}$. It assumes that the conditional probability density functions $p(x|y = 0)$ and $p(x|y = 1)$ are both normal distributions with parameters of covariance and mean for each class, i.e., $(\Sigma_0, \mu_0)$ and $(\Sigma_1, \mu_1)$ (Fisher, 1936). LDA computes $(\Sigma_0, \mu_0)$ and $(\Sigma_1, \mu_1)$ given the training data and learns a classifier based on them.

Most LDA methods make the further simplifying homoscedasticity assumption that the class covariances are identical, i.e., $\Sigma_0 = \Sigma_1 = \Sigma$. Thus, the differences between different classes are in only their means $\mu_i$'s. This assumption is particularly useful for continual learning as the system does not have to save one covariance matrix for each class, which can consume a huge amount of memory as more classes are learned. Otherwise, the space required to save covariance matrices for all classes will be huge. LDA also makes it possible without using any replay data. OpenLD uses multiclass LDA, which allows any number of classes to be classified (Rao, 1948).

### 3.2   OPENLD METHOD

The proposed method OpenLD uses a *pre-trained model* (which is frozen throughout), *a continual OOD detection method*, and the *linear discriminant analysis* (LDA) method to solve the OWCL problem. The pre-trained model will be described in the experiment section. The proposed system OpenLD consists of the following steps.

**1. Building the Initial Model $M$.** OpenLD uses the pre-trained model $f$ to provide features for the input samples, which is used by LDA to build a model or classifier $M$ using the initial classes $C$. As mentioned earlier, LDA's classifier building is based on a mean $\mu_i$ for each class $i \in C$ and a single

shared covariance matrix $\Sigma$ across all classes. Thus, it produces the shared $\Sigma$ and a separate mean $\mu_i$ for each class $i \in C$. The resulting model $M$ is deployed in its application (Figure 1).

**2. Post-deployment Continual Learning.** After deployment, it continues to learn, which will update $M$ after new classes are incrementally learned and $M$ becomes $M^+$. In the continual learning process, $\Sigma$ remains unchanged or frozen and it is also used by the newly detected classes.

Each iteration of post-deployment continual learning has two sub-steps.

**2.1. Continual OOD Detection and Classification.** OpenLD uses $M^+$ to detect whether each income sample $x$ in the online stream is an OOD sample. If not, it is classified to its class (see Figure 1). Note that $M^+$ is $M$ initially. OpenLD employs the covariance matrix $\Sigma$ and the $\mu_i$'s for all classes encountered or seen so far to perform the tasks.

At a steady state, the set of all classes that the system has encountered is $C^A = C \cup C^L \cup C^E$, where $C$ is the initial set of classes learned in $M$, $C^L$ is a set of new classes that have appeared after model deployment and are *well learned* after seeing a good number of training instances, and $C^E$ is a set of emerging new classes that have been seen but are *not well learned yet*, i.e., not enough labeled training data have been seen to well learn the classes. **Well learned** means that the mean of the class has stabilized or converged, i.e., it does not change much after more samples are added. A class becomes well-learned if $||\mu_{new} - \mu_{old}|| < th$, where $\mu_{old}$ and $\mu_{new}$ denote the mean of the class before and after being updated by a sample of the class, and $th$ is a threshold. We denote $C^+ = C \cup C^L$ as the set of well-learned classes so far and $C^N = C^L \cup C^E$ as the set of all new classes seen after the deployment of $M$. With incremental learning, $M$ becomes $M^+$, covering all classes in $C^A$.

What is important here is that OOD detection not only uses the classes in $C^+$ but also leverages the covariance $\Sigma$ and the current un-converged means of the classes in $C^E$ to detect OOD samples belonging to $C^E$ and other new classes. To our knowledge, no existing method does that. This is advantageous because a new sample may be similar to a class in $C^E$, which makes OOD detection more effective. In Sec. 3.3, we discuss the OOD detection methods used in our OpenLD. If a test sample $x$ is near a class in $C^E$, it is also considered an OOD sample.

**2.2 Continual Learning - *class-incremental learning* (CIL).** Here the continual learning setting is CIL, which incrementally learns more and more new classes. Specifically, OpenLD learns each detected novel instance $x$ after obtaining its class label by asking a human user or another knowledgeable agent. If $x$ is assigned a class label in $C$, do nothing (i.e., no learning). If $x$ is assigned a class label $c_i$ in $C^N$, the current model $M^+$ is updated by updating the mean $\mu_i$ of the class $c_i$ (covariance matrix $\Sigma$ is not changed) as

$$\mu_i \leftarrow \frac{n_i \mu_i + z}{n_i + 1}, \tag{1}$$

where $z$ is the feature $f(x)$ obtained from the pre-training feature extractor $f$ and $n_i$ is the number samples seen so far in class $i$.

This approach has two desirable properties.

(1) OpenLD has no catastrophic forgetting (CF) during OWCL after model deployment because we use the frozen pre-trained feature extractor, a fixed and shared covariance $\Sigma$, and the running means for each class are independent of those of other classes. Thus, there is no interference across classes.

(2) Again, due to the sharing of covariance matrix $\Sigma$ by all old and new classes, we achieve strong learning results with a small number of examples because, for each detected new class, OpenLD only updates its mean based on the identified samples of the class.

### 3.3 OOD DETECTION METHODS

Since OpenLD produces a shared covariance $\Sigma$ and one mean $\mu_i$ for each class $i$, we can naturally use $\Sigma$ and $\mu_i$ related OOD detection methods, i.e., *Mahalanobis distance* (MD), relative *Mahalanobis distance* (RMD), and *distance to mean*. Each of these methods produces a confidence score using all classes $k \in C^+$ for the given feature vector $z = f(x)$, where $x$ is the input. If the confidence score remains below a *threshold* level, or it belongs to any class in $C^E$, that input is marked as OOD. Note that apart from these methods, there are numerous existing OOD detection methods

(see Sec. 2). However, since our approach does not train a neural network for prediction in LDA, most of those methods are not suitable for use in OpenLD.

### 3.3.1 MAHALANOBIS DISTANCE (MD)

Mahalanobis distance (McLachlan, 1999) measures the distance between a feature and a distribution using the class mean vector $\mu$ and the covariance $\Sigma$. Note that, each class mean $\mu_i$ and covariance $\Sigma$ for the data used in building the initial model $M$ are estimated as: $\mu_i = \frac{1}{N_i} \sum_{k:y_k=i} z_k$ and $\Sigma = \frac{1}{N} \sum_{i \in C} \sum_{k:y_k=i} (z_k - \mu_i)(z_k - \mu_i)^T$, where $N$ denotes the number of samples, $N_i$ denotes the number of samples of class $i$, and $z_k$ is the feature of input sample $x_k$ obtained from the pre-trained model, i.e., $z_k = f(x_k)$. $\Sigma$ is the same for new classes, while $\mu_i$ for the new classes are incrementally computed using Eq. 1.

For the given $z = f(x)$ of a test sample $x$, we can compute MD as,

$$MD_i(z; \mu_i, \Sigma) = \sqrt{(z - \mu_i)^T \Sigma^{-1}(z - \mu_i)} \tag{2}$$

where $\Sigma^{-1}$ is the inverse of covariance matrix. The confidence score $c$ is described as,

$$c(z) = \max_{i \in C^+} \{1/MD_i(z; \mu_i, \Sigma)\} \tag{3}$$

### 3.3.2 RELATIVE MAHALANOBIS DISTANCE (RMD)

As noted in (Ren et al., 2021), MD has some limitations regarding the detection of OOD data and they proposed RMD by applying a simple addition to MD. It computes an additional mean $\mu_C = \frac{1}{N} \sum_{k=1}^{N} z_k$ and covariance $\Sigma_C = \frac{1}{N} \sum_{k=1}^{N} (z_k - \mu_C)(z_k - \mu_C)^T$, which, in our case, are only calculated based on the initial data with $C$ classes used in building model $M$. RMD is computed as,

$$RMD_i(z; \mu_i, \Sigma, \mu_C, \Sigma_C) = MD_i(z; \mu_i, \Sigma) - MD_A(z; \mu_C, \Sigma_C) \tag{4}$$

where $MD_A(z; \mu_C, \Sigma) = \sqrt{(z - \mu_C)^T \Sigma_C^{-1}(z - \mu_C)}$. The confidence score is (Ren et al., 2021)

$$c(z) = \max_{i \in C^+} \{-RMD_i(z; \mu_i, \Sigma, \mu_C, \Sigma_C)\} \tag{5}$$

### 3.3.3 DISTANCE TO MEAN

This method uses the norm of the difference between feature $z$ and class mean $\mu_i$. Given these vectors, a confidence score is computed as,

$$c(z) = \max_{i \in C^+} \{1/(||z - \mu_i||)\} \tag{6}$$

## 4 EXPERIMENTAL EVALUATION

### 4.1 DATASETS, COMPARED METHODS, PRE-TRAINED MODEL, AND IMPLEMENTATION

**Datasets.** We utilize three well-known continual learning benchmark datasets in our experiments. **(1) CIFAR-10** (Krizhevsky et al., 2009): This dataset comprises 60,000 color images with dimensions of 32x32 pixels. It includes 50,000 training images and 10,000 testing images, distributed evenly across 10 classes. **(2) CIFAR-100** (Krizhevsky et al., 2009): This dataset contains 50,000 training images (500 per class) and 10,000 testing images (100 per class) of 100 classes, all of which are colored and sized at 32x32 pixels. **(3) TinyImageNet** (Le & Yang, 2015): This dataset has 200 classes, each with 500 training images of 64x64 pixels. The validation set includes 50 images per class. Since the test data labels are unavailable, we use the validation set for testing.

**– ID Class Set and OOD Class Set:** For each experiment using a dataset, we divide the classes in the dataset into an equal number of ID (in-distribution) classes and OOD classes. The **ID class set** is used for building the initial model $M$ for deployment, while the **OOD class set** is used in incremental learning after model deployment.

**– ID Train Set and ID+OOD APP Set:** We further divide the training set of each class in the ID class set into *ID Train set* (80%) and *ID APP set* (20%). We do the same for the OOD class set. *ID+OOD APP set* includes the data from both the ID and OOD APP sets. The ID+OOD APP set simulates the application (APP) data from a real-life data stream that needs to be classified.

For CIFAR-100 and TinyImageNet, each class has 500 samples in its original training set. After the split, the ID Train set has 400 samples and the ID+OOD APP set has 100 samples per class. CIFAR-10 has 5000 samples per class in its original training set, but since we want to simulate the situation where the system does not ask the human user too many questions, we selected 4000 samples per class for the ID Train set and only 100 samples per class for the ID+OOD APP set.

**– Pre- and Post-Deployment:** In pre-deployment, we perform joint training using LDA and a pre-trained model (see below) to build the model $M$ using only the ID Train set. ID+OOD APP set is used only post-deployment.

**Compared Methods.** Although there are several related papers (Bendale & Boult, 2015; Gummadi et al., 2022; Rios et al., 2022; Roy et al., 2022), as discussed in Sec 2, no existing system can perform OWCL after model deployment as proposed in this paper. However, we found that the recent continual learning system TPL (Lin et al., 2024) can be adapted for OWCL as it is based on OOD detection. We also create variations of the proposed method OpenLD as **ablation experiments**.

**– TPL:** TPL (Lin et al., 2024) is a state-of-the-art CIL system for traditional continual learning. Since each task in TPL is learned as an OOD detection model and the OOD score for each task is used to predict the task-id of each test sample, it can naturally be adapted for OWCL. Our OpenLD also uses the same pre-trained model as TPL. To suit OWCL, we made it learn in one epoch with a batch size of 1 to simulate learning from a data stream like OpenLD in post-deployment. ID Train set is used to learn ID classes to build the initial model $M$ as the first task, and OOD APP set is used to learn the OOD classes incrementally. The setting assumes that TPL can do perfect OOD detection (thus ID APP set is not used). We did not use TPL to do OOD detection as it is hard to set an OOD score threshold because it depends on two factors. However, even in this ideal case for TPL, it performs markedly poorer than OpenLD.

**– OpenLD[X]:** This creates three variations of OpenLD, using three OOD detection methods (X), i.e., MD, RMD, or Mean for OOD detection (see Section 3.3).

**– OpenLD[X](-$C^E$):** This also creates three variations of OpenLD *without* leveraging the un-converged means of the classes in $C^E$ to help detect OOD samples. These are for **ablation** experiments and they will show that using $C^E$ in OOD detection is very helpful.

**Two upper-bound methods** are also created, which are not continual learning methods.

**– Joint LDA:** This method applies LDA to learn a classifier using the data from all classes together or jointly, i.e., ID Train set and OOD APP set. ID APP set is not used as there is no post-deployment learning or OOD detection in this setting. This method gives the upper bound results of LDA.

**– Joint Fine-tune:** This joint method fine-tunes the pre-trained model using ID Train set and OOD APP set. We used the AdamW optimizer with a learning rate of 0.0005, batch size of 256, and trained the model for 20 epochs, which is sufficient for convergence.

**Pre-trained Models.** For feature extraction, our primary pre-trained model is DeiT-S/16 (Touvron et al., 2021) as in (Kim et al., 2022a). To prevent information leaks between the pre-training and continual learning phases, 389 classes in ImageNet, which are similar to the classes in CIFAR-10, CIFAR-100, and TinyImageNet, were excluded and pre-training was performed with the remaining 611 classes. In Section 4.3.1, we will also see the results of using several other pre-trained models.

**Implementation and Resource Usage.** We used the LDA implementation in (Hayes & Kanan, 2020) and the pre-trained model in (Kim et al., 2022a), which is a DeiT-S/16 Transformer (Touvron et al., 2021), also used in TPL Lin et al. (2024). We run the experiments on a machine with an AMD EPYC 7502 32-Core Processor and NVIDIA RTX A6000 GPU. Each experiment requires approximately 6 GB GPU memory and takes an average of 10 minutes.

**OOD Detection and $C^L$ Thresholds.** In each setup, confidence scores for OOD detection are computed using three alternative methods, RMD, MD, and Mean, and an image input is considered OOD if its confidence score is less than a certain threshold. These thresholds are set to 0, 0.07, and

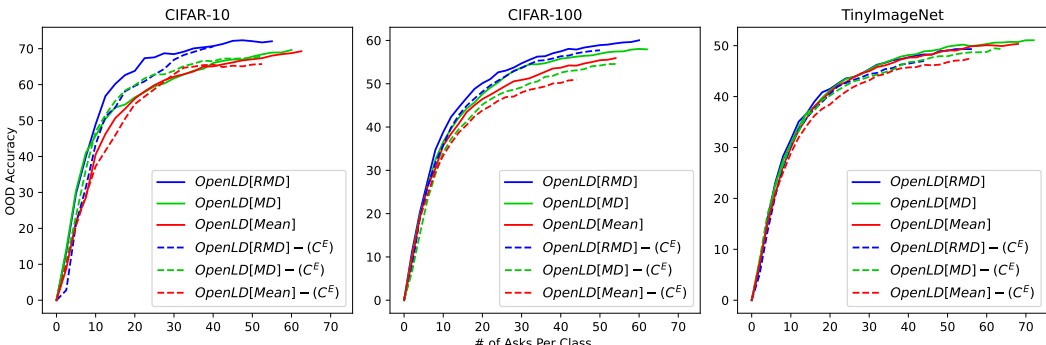

Figure 2: {OOD class classification accuracy}-to-{number of asks} plots for OpenLD's *Random ID+OOD APP Data Arrival* experiment setup.

0.08 for RMD, MD, and Mean, respectively. They are chosen empirically to ensure that the precision and recall for the OOD data are similar for each method. The convergence threshold *th* for $C^L$ is set to 0.1 empirically. We tested values between 0.1 and 10. As the threshold increases, accuracy and the number of asks decrease as expected, with a maximum reduction of 2.9% in accuracy and 46.1% in the number of asks. We chose these threshold values, as the system not only wants to achieve good accuracy but also to have a good balance between accuracy and the number of asks.

## 4.2 Two experimental setups and evaluation measures

**Setup 1) Random ID+OOD APP Data Arrival.** This is ***our main setup*** as it reflects a normal real-life application scenario. In this setup, after model deployment, the samples in the ID+OOD APP set arrive randomly in a data stream. The current model $M^+$ classifies each to its class or detects it as OOD. For each detected OOD sample (which may be correct or wrong), the proposed system OpenLD asks the human user for its class label, and then OpenLD incremental/continual learning is performed by updating its class mean. Clearly, in our experiment, ***no human user*** is involved. The system just uses the class label of the sample in the original data.

**Setup 2) Class-Incremental OOD APP Data Arrival.** This setup is for scenarios where an AI agent keeps going to completely new environments. This is very unlikely. We use it to show the robustness of our approach. Each environment has a set of new OOD classes as a new task. This is similar to class incremental learning (CIL). In our case, we divide the classes in the OOD class set evenly into 5 tasks (5 environments). Each task is created to include one-fifth of random samples from the ID APP set and all samples in the ODD APP set of the classes in the task. The OOD samples from one task finish before the data from the next task arrive. Acquiring the OOD sample class label and updating the class means for incremental/continual learning are the same as above.

**Evaluation Measures.** We conduct two types of evaluations.

(i). Draw a curve to show how the accuracy changes as the number of asks increases. ***Setup 1)***: For CIFAR-10, for every 25 asks (i.e., detected OOD samples, some of which may be true OOD samples and some of which may be not), accuracy is computed using the test data from all classes. For CIFAR-100 and TinyImageNet, for every 100 asks, accuracy is computed as these two datasets have many more OOD classes. ***Setup 2):*** It is similar. For every 25 asks for CIFAR-10 and every 100 asks for CIFAR-100 and TinyImageNet, accuracy is computed using the test data of the classes that the model has seen so far.

(ii). Final test **accuracy** after the system has seen all the data in ID+OOD APP set. Final **F score** for OOD detection also after the system has seen all the data in ID+OOD APP set.

## 4.3 Results and analysis

We report the results of the two experiment setups. Due to space limits, we report the results for the OOD classes in the OOD class set as we are mainly interested in learning after model deployment in the main text of the paper. The full results (including both ID classes and OOD classes) are given in Appendices A.1 and A.2.

Table 1: Performance comparison of OpenLD using different OOD score methods for CIFAR-10, CIFAR-100 and TinyImageNet datasets in the *Random ID+OOD APP Data Arrival* experiment setup. F score gives the OOD detection performance. Joint methods and TPL baseline do not have F score as they do not do OOD detection. Note that the standard deviations are very small because LDA is deterministic. Each experiment was run with 3 random seeds.

| | CIFAR-10 | | CIFAR-100 | | TinyImageNet | | Average | |
|---|---|---|---|---|---|---|---|---|
| Methods | F Score | Accuracy | F Score | Accuracy | F Score | Accuracy | F Score | Accuracy |
| *Joint LDA(Upper-bound)* | | $82.89_{\pm 0.00}$ | | $65.80_{\pm 0.00}$ | | $56.69_{\pm 0.00}$ | | $68.46$ |
| *Joint Fine-tune(Upper-bound)* | | $82.60_{\pm 0.81}$ | | $75.81_{\pm 0.32}$ | | $64.88_{\pm 0.57}$ | | $74.86$ |
| TPL | | $77.56_{\pm 0.79}$ | | $63.17_{\pm 0.70}$ | | $49.88_{\pm 0.08}$ | | $63.53$ |
| OpenLD[Mean]($-C^E$) | $60.85_{\pm 0.00}$ | $78.36_{\pm 0.00}$ | $57.55_{\pm 0.00}$ | $60.94_{\pm 0.00}$ | $60.23_{\pm 0.00}$ | $54.81_{\pm 0.00}$ | $59.54$ | $64.70$ |
| OpenLD[MD]($-C^E$) | $65.58_{\pm 0.00}$ | $79.21_{\pm 0.01}$ | $61.89_{\pm 0.00}$ | $62.63_{\pm 0.01}$ | $62.96_{\pm 0.00}$ | $55.29_{\pm 0.00}$ | $63.47$ | $65.71$ |
| OpenLD[RMD]($-C^E$) | $58.74_{\pm 0.00}$ | $80.76_{\pm 0.00}$ | $68.14_{\pm 0.03}$ | $63.53_{\pm 0.02}$ | $60.26_{\pm 0.00}$ | $54.23_{\pm 0.00}$ | $62.38$ | $66.17$ |
| OpenLD[Mean] | $70.05_{\pm 0.31}$ | $79.94_{\pm 0.07}$ | $68.14_{\pm 0.13}$ | $63.08_{\pm 0.05}$ | $68.25_{\pm 0.13}$ | $55.73_{\pm 0.09}$ | $70.48$ | $66.25$ |
| OpenLD[MD] | $72.02_{\pm 0.54}$ | $80.12_{\pm 0.09}$ | $69.32_{\pm 0.09}$ | $63.93_{\pm 0.05}$ | $68.67_{\pm 0.17}$ | $55.99_{\pm 0.04}$ | $70.00$ | $66.68$ |
| OpenLD[RMD] | $77.31_{\pm 1.02}$ | $81.15_{\pm 0.10}$ | $76.01_{\pm 0.39}$ | $64.30_{\pm 0.19}$ | $73.77_{\pm 0.23}$ | $55.27_{\pm 0.03}$ | **$75.69$** | **$66.90$** |

### 4.3.1 SETUP 1): RANDOM ID AND OOD APP DATA ARRIVAL

As mentioned in Sec. 4.2, this is our main experiment setup as it is the most realistic scenario. We draw the curve for each dataset in Figure 2 to trace the accuracy changes of the OOD classes with the number of asks. Dashed lines represent OpenLD variants without using $C^E$ in OOD detection. The solid lines represent OpenLD variants using $C^E$ in OOD detection. The lengths of the lines vary depending on when that method stopped asking. The curves show that our method OpenLD consistently outperforms the methods using traditional OOD detection techniques (without using $C^E$) in terms of accuracy for OOD classes in the OOD class set. That is, using the same OOD detection technique, when we include the classes in $C^E$ in detecting OOD samples as in our proposed method OpenLD, accuracy increases throughout the open-world continual learning stage. The plots for the total accuracy (including both ID classes and OOD classes) against the number of asks are given in Figure 4 in Appendix A.1, where we will see the same trend.

In addition, Table 1 shows the OOD detection F score, and accuracy of the combined ID and OOD classes after all data in the ID+OOD APP set are seen. OpenLD variants (which use $C^E$) give significantly higher F scores than their corresponding variants without using $C^E$. The final accuracy is also better. Note that the accuracy improvement is not large as the test results are obtained after the system sees all data, at which time the means of the OOD classes have mostly converged. We also observe that the state-of-the-art CIL system TPL, which also does feature learning, is markedly poorer even in its ideal situation, i.e., OOD detection is perfectly done without any errors.

**Using Different Pre-Trained Models.** We also examined the OpenLD's performance using different pre-trained models (see Table 2). DeiT-S/16-ImageNet1k (Touvron et al., 2021) and ViT-B/16-ImageNet21k (Dosovitskiy et al., 2020), trained using supervised data, give better performance as the classes of their pre-training overlap with the classes used in our experimental data, which causes *information leaks*. ViT-S/16-DINO (Caron et al., 2021), and ViT-B/16-SAM (Kirillov et al., 2023), trained in a self-supervised manner, gave poorer results, which is expected. ViT-B/16-DINO (Caron et al., 2021), although also trained in a self-supervised manner, is better in accuracy because it is a much larger model, but is weak in OOD detection. These results show that OpenLD gives good performance with current pre-trained models. As new and more powerful pre-trained models appear constantly, the results will improve further and there will be less and less need to learn new features or fine-tune the pre-trained models, which can cause catastrophic forgetting (CF).

**Efficiency.** Since OpenLD uses the statistical techniques LDA in learning and MD for OOD detection, it is much more efficient than deep learning. In the post-deployment stage, the system performs only OOD detection and updating of the mean of each new class (no training is involved), which takes almost no time (less than 15 milliseconds), because the features are already learned in the pre-trained model. This makes OpenLD especially suitable for OWCL and real-time processing.

### 4.3.2 SETUP 2): CLASS-INCREMENTAL OOD DATA ARRIVAL

Since this setup consists of multiple tasks learned incrementally, we created a figure for each task. Figure 3 shows the change in the accuracy of OOD classes in the OOD class set within each task.

Table 2: Performance comparison of OpenLD using different pre-trained models. All experiments use RMD as the OOD score method. Note that in the name of each model, S means *small*, B means *base*, and 16 means that each image is divided into 16 patches in pre-training.

| Models | CIFAR-10 | | CIFAR-100 | | TinyImageNet | | Average | |
|---|---|---|---|---|---|---|---|---|
| | F Score | Accuracy | F Score | Accuracy | F Score | Accuracy | F Score | Accuracy |
| DeiT-S/16 (Kim et al., 2022a) | $77.31_{\pm1.02}$ | $81.15_{\pm0.10}$ | $76.01_{\pm0.39}$ | $64.30_{\pm0.19}$ | $73.77_{\pm0.23}$ | $55.27_{\pm0.03}$ | 75.69 | 66.90 |
| DeiT-S/16-ImageNet1k | $80.49_{\pm0.64}$ | $84.14_{\pm0.12}$ | $75.96_{\pm0.20}$ | $66.81_{\pm0.19}$ | $78.93_{\pm0.09}$ | $68.75_{\pm0.11}$ | 78.46 | 73.23 |
| ViT-B/16-ImageNet21k | $85.20_{\pm0.39}$ | $85.53_{\pm0.02}$ | $78.16_{\pm0.08}$ | $66.19_{\pm0.05}$ | $78.92_{\pm0.08}$ | $64.01_{\pm0.03}$ | 80.76 | 71.91 |
| ViT-S/16-DINO | $23.92_{\pm4.31}$ | $50.29_{\pm0.72}$ | $13.27_{\pm0.52}$ | $40.88_{\pm0.25}$ | $8.29_{\pm1.03}$ | $35.05_{\pm0.39}$ | 15.16 | 42.07 |
| ViT-B/16-DINO | $61.40_{\pm0.91}$ | $79.48_{\pm1.15}$ | $59.56_{\pm0.33}$ | $64.88_{\pm0.28}$ | $53.96_{\pm0.65}$ | $58.53_{\pm0.62}$ | 58.30 | 67.63 |
| ViT-B/16-SAM | $68.88_{\pm0.89}$ | $73.01_{\pm0.41}$ | $68.76_{\pm0.10}$ | $56.27_{\pm0.06}$ | $56.57_{\pm0.54}$ | $37.87_{\pm0.14}$ | 64.73 | 55.71 |

Figure 3: {OOD class classification accuracy}-to-{number of asks} plots for the OpenLD's *Class-Incremental OOD APP Data Arrival* experiment setup.

Similar to the previous setup, OpenLD (which uses $C^E$) demonstrates better performance for each dataset than without using $C^E$ in OOD detection. Due to space limits, the total accuracy plots and the results for OOD detection F score and the final accuracy after seeing all the data are given in Figure 5 and Table 3 , in Appendix A.2. They all show the effectiveness of OpenLD.

## 5 CONCLUSION

This paper proposed the setting of open-world continual learning (OWCL) after model deployment. The OWCL setting is getting more and more important as more and more AI agents are deployed in real-life applications. It is highly desirable that these agents can learn continually after deployment to improve themselves and become more and more knowledgeable over time. This paper also proposed a method (called OpenLD) based on LDA and a pre-trained model with several novel techniques to improve OOD detection in the OWCL process and to learn new classes easily by only updating class means, which has no catastrophic forgetting (CF). Experiment results have demonstrated the effectiveness of the proposed method OpenLD.

**Limitations.** (1) The pre-trained model is frozen, meaning no feature learning from new data, although OpenLD is better than TPL that does feature learning. (2) The type of OOD detection method is limited to distance-based methods because LDA only generates a shared covariance and a mean for each class. In our future work, we will try to overcome these limitations.

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

# A APPENDIX

This section contains additional results for each experimental setup.

## A.1 ADDITIONAL RESULTS FOR THE RANDOM ID+OOD APP DATA ARRIVAL SETUP

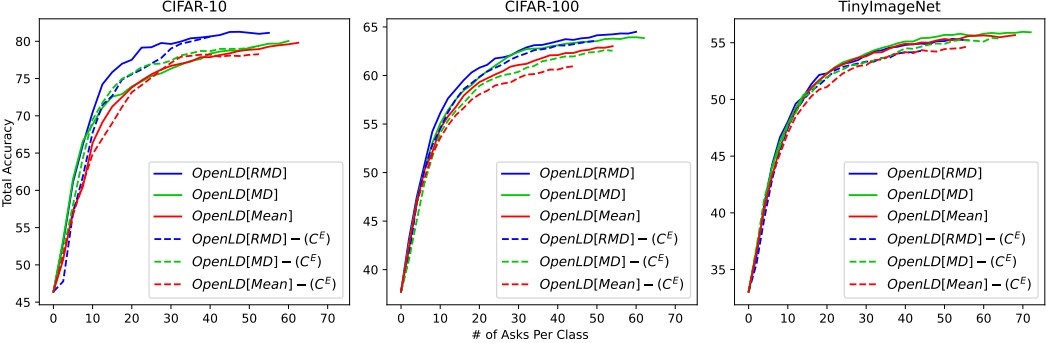

Figure 4: {Total accuracy}-to-{number of asks} plots for the OpenLD's *Random ID+OOD APP Data Arrival* experiment setup. Each accuracy is computed using the test sets of all ID classes and all OOD classes.

## A.2 ADDITIONAL RESULTS FOR THE CLASS-INCREMENTAL OOD APP DATA ARRIVAL SETUP

Note that we did not do the experiments using the other pre-trained models as this setup is not our main experiment setup because it is very unlikely to happen in practice. We use it to show the robustness of our approach.

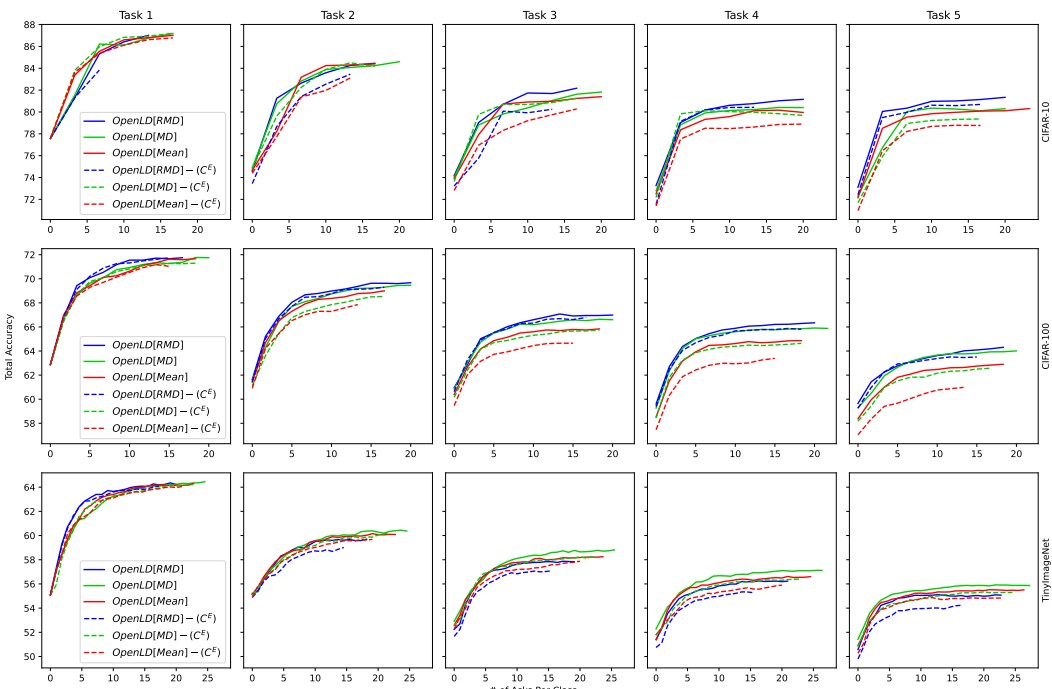

Figure 5: {Total accuracy}-to-{number of asks} plots for the OpenLD's *Class-Incremental OOD APP Data Arrival* experiment setup. The accuracy of each task is computed using all ID classes and OOD classes the model has seen so far.

Table 3: Performance comparison of OpenLD using different OOD score methods for CIFAR-10, CIFAR-100 and TinyImageNet datasets in the *Class-Incremental OOD APP Data Arrival* experiment setup. F score gives the OOD detection performance in the open world continual learning process. Note that TPL does not have F score values as we assume that its OOD detection is perfect.

| Methods | CIFAR-10 | | CIFAR-100 | | TinyImageNet | | Average | |
|---|---|---|---|---|---|---|---|---|
| | F Score | Accuracy | F Score | Accuracy | F Score | Accuracy | F Score | Accuracy |
| TPL | | $79.06_{\pm1.07}$ | | $61.68_{\pm1.44}$ | | $48.18_{\pm0.02}$ | | 62.97 |
| OpenLD[Mean]($-C^E$) | $61.93_{\pm0.00}$ | $78.77_{\pm0.00}$ | $57.55_{\pm0.00}$ | $60.94_{\pm0.00}$ | $60.23_{\pm0.00}$ | $54.81_{\pm0.00}$ | 59.90 | 64.84 |
| OpenLD[MD]($-C^E$) | $66.22_{\pm0.00}$ | $79.42_{\pm0.00}$ | $61.89_{\pm0.00}$ | $62.63_{\pm0.01}$ | $62.96_{\pm0.00}$ | $55.29_{\pm0.00}$ | 63.69 | 65.78 |
| OpenLD[RMD]($-C^E$) | $56.89_{\pm0.00}$ | $80.69_{\pm0.01}$ | $68.04_{\pm0.11}$ | $63.56_{\pm0.05}$ | $60.26_{\pm0.00}$ | $54.23_{\pm0.00}$ | 61.73 | 66.16 |
| OpenLD[Mean] | $70.19_{\pm0.36}$ | $80.30_{\pm0.03}$ | $67.98_{\pm0.16}$ | $62.77_{\pm0.05}$ | $68.51_{\pm0.24}$ | $55.62_{\pm0.11}$ | 68.89 | 66.23 |
| OpenLD[MD] | $72.80_{\pm0.65}$ | $80.28_{\pm0.18}$ | $69.76_{\pm0.25}$ | $63.85_{\pm0.13}$ | $68.97_{\pm0.03}$ | $56.01_{\pm0.08}$ | 70.51 | 66.71 |
| OpenLD[RMD] | $74.72_{\pm1.10}$ | $81.40_{\pm0.03}$ | $75.60_{\pm0.38}$ | $64.07_{\pm0.18}$ | $73.62_{\pm0.18}$ | $55.15_{\pm0.26}$ | **74.64** | **66.87** |

