# OpenReview forum: "Continual Learning After Model Deployment"
_ICLR.cc/2025/Conference — ICLR 2025 Conference Withdrawn Submission_

### Official Review · Reviewer_NFvs · 2024-10-16

**Soundness:** 2
**Presentation:** 2
**Contribution:** 1
**Rating:** 3
**Confidence:** 3

**Summary:**

Overall, this paper constructs a new continual learning setting named open-world continual learning. After that, it proposes a novel method called OpenLD under this setting.

**Strengths:**

The problem that this paper aims to handle is important, as continual learning does should be expected to happen in an open-set scenario.

The paper achieves good experimental results.

**Weaknesses:**

(See the questions section below)

**Questions:**

Overall, I believe that this paper is currently not ready to be accepted. Below are my concerns.

1. I am kind of confused over the difference between the proposed setting and the simple combination of OOD detection and few-shot class incremental learning. Considering this, I am curious that, why simply combining an existing OOD method and an existing few-shot class incremental learning method cannot handle the proposed problem.

2. Meanwhile, I am also a bit confused over the difference between OOD detection and continual OOD detection. This is because, while the authors seem to try to highlight that OOD detection can face many challenges in their proposed setting, they seem to finally just use existing OOD methods to perform OOD detection in their setting. Thus, does this mean that it is just the most typical OOD detection that is performed in the proposed framework?

3. I am confused over the practicity of the proposed setting. Specifically, continual learning is also known as lifelong learning. Yet, in the proposed setting, if I am not wrong, it seems that a person is always required to be involved to annotate every detected OOD data. This is quite strange and non-realistic to me.

4. Meanwhile, the paper seems to heavily base their method on an assumption that all classes share the same covariance. I believe that this can be a non-realistic assumption. Specifically, it is very natural that some classes hold a large intra-variance than other classes. The constraint that all classes must have the same covariance is thus a very strict assumption from my perspective.

In summary, in light of the above, I believe that this submission is not ready for being published in its current form.

---

### Official Review · Reviewer_vex5 · 2024-10-29

**Soundness:** 2
**Presentation:** 2
**Contribution:** 2
**Rating:** 3
**Confidence:** 4

**Summary:**

The paper proposes an open-world continual learning setting where the model can detect OOD classes and learn new classes in a class-incremental learning setup without fine-tuning the pre-trained model. The paper also proposes a realistic setup for class-incremental learning where the incremental data contains a mix of in-distribution and out-of-distribution classes. The proposed method uses distance metrics to detect OOD classes and uses LDA for class-incremental learning. The results are shown on three datasets including CIFAR-10, CIFAR-100 and TinyImageNet.

**Strengths:**

1. The author identifies important and realistic challenges in continual learning. The proposed problem setting of open-world continual learning is relevant and realistic compared to previously studied class-incremental learning settings where new class data arrives in pure chunks of new classes. Combining OOD detection with continual learning in a realistic setup is a step in the correct direction.

2. The proposed approach is quite neat where both OOD class detection and incremental learning can be performed using similar distance-based approaches. The results show that the proposed openLD method is able to handle the upcoming new OOD classes without losing performance obtained using the LDA classifier.

3. Related work is well-written and provides a good summary of relevant work on continual learning and OOD detection.

**Weaknesses:**

The problem setup is quite relevant for the field, but the application of the method in this work has some limitations.
1. Firstly, the paper title is a bit deceiving since model deployment usually refers to a model without model weights access, which is assumed to be available in this work.
2. The pre-trained is only used for feature extraction and not for fine-tuning using the ID train set before deployment. Since there is a big gap in performance using the LDA classifier and fine-tuned model, it would make sense to start with the strongest possible model on ID classes. Additionally, it is not realistic to ignore or throw away the new upcoming ID APP data “after deployment”.
3. The method keeps the pre-trained model ‘frozen’ all the time and only performs continual learning on the obtained features. This setting is highly limiting for the performance of the model and does not reflect a realistic continual learning setting when model weights are accessible.
4. Missing comparison with similar baselines like nearest neighbor approach or nearest centroid approach or prototypical network. The benefits of using LDA are not motivated.
5. In Table 1, the difference between the OpenLD and Joint Fine-tuning upper bound grows as the model is scaled to a dataset with a larger number of classes, showing the approach is not scalable and relies heavily on the extracted features from the frozen model.
6. The presentation of the paper can be improved. The captions are not self-complete and the text contains copied sentences. For example, the Figure 1 caption, does not explain the figure. Table 1 caption does not say which performance is shown in the table. The same text describing the OWCL setting is repeated from the introduction to Section 2.1 reducing the quality of the paper.

Minor comments
1. Line 454 claims that OpenLD consistently outperforms the methods without using C^E. However, this is not true for CIFAR-10 shown in Figure 2.
2. In Line 471, can the authors explain why VIT-S/16 DINO and VIT-B/16 SAM are expected to show poorer results?
3. In Line 472, although VIT-B/16 DINO is as big as VIT-B/16 SAM, why is it expected to perform better?

**Questions:**

1. Is the pre-trained model fine-tuned with the samples from the ID train set before deployment? If not, please reason about it. This is important to know because there is a significant gap between fine-tuned joint-training performance and Joint LDA performance.
2. Since the classes of CIFAR and miniImageNet datasets are removed from the pre-training dataset, how is it ensured that features extracted from the pre-trained frozen model generalize for new classes?
3. Why did authors use LDA and not other distance-based methods? Please include comparisons with other off-the-shelf classification methods like nearest centroid-based classification or kNN.
4. The paper requires improved presentation and stronger reasoning behind different design choices with ablations and comparisons. The proposed model should be compared with the strongest fine-tuning-based baseline setup to make claims about the removal of catastrophic forgetting.

---

### Official Review · Reviewer_ipVQ · 2024-10-30

**Soundness:** 4
**Presentation:** 3
**Contribution:** 3
**Rating:** 6
**Confidence:** 4

**Summary:**

This paper proposes a novel setting called Open-World Continual Learning (OWCL), which addresses the real-world scenario where a model must continue learning after deployment to handle new, unseen objects (out-of-distribution, OOD). The proposed method, OpenLD, is based on Linear Discriminant Analysis (LDA) combined with a pre-trained model to enable efficient OOD detection and incremental learning without catastrophic forgetting. Experimental results on benchmark datasets (CIFAR-10, CIFAR-100, TinyImageNet) demonstrate that OpenLD outperforms existing methods in both OOD detection and continual learning after model deployment.

**Strengths:**

1.	I think this is a very meaningful topic.It introduces Open-World Continual Learning (OWCL), which is significant for AI models operating in real-world environments. OWCL allows the model to continuously adapt after deployment, enhancing its ability to autonomously acquire new knowledge.
2.	The OpenLD effectively combines Linear Discriminant Analysis (LDA) with a pre-trained model to handle OOD detection and incremental learning. This approach avoids catastrophic forgetting by using a shared covariance matrix and updating class means incrementally.
3.	The experimental results on standard benchmark datasets (CIFAR-10, CIFAR-100, TinyImageNet) demonstrate that OpenLD performs better in terms of both accuracy and robustness compared to existing methods. This validates its effectiveness in an open-world continual learning scenario.

**Weaknesses:**

1.	OpenLD relies too much on pre-trained models, which makes it unable to learn new features from new data in existing categories after deployment. Will this limit the recognition of known categories?
2.	The OpenLD method uses a shared covariance matrix to handle all categories, which can become problematic as the number of categories increases significantly. A shared covariance matrix can result in reduced accuracy or increased computational burden when managing a large number of categories.
3.	I think your article is lacking in the methodological explanation, such as why Marhalanobis distance is used, what are the advantages of Marhalanobis distance, and whether there are theoretical or experimental advantages over other distances.
4.	I think you can include as many comparison methods as possible. Although this is a very novel question, similar methods can be compared, and your table needs to be beautified.

**Questions:**

Have you considered combining OpenLD with other state-of-the-art OOD detection approaches, such as those using neural network uncertainty or ensemble methods, to improve robustness?

---

### Official Review · Reviewer_XK34 · 2024-10-31

**Soundness:** 2
**Presentation:** 1
**Contribution:** 2
**Rating:** 3
**Confidence:** 5

**Summary:**

This paper introduces OpenLD, a method for open-world continual learning (OWCL), where models deployed in real-world environments encounter novel, out-of-distribution (OOD) objects. OWCL combines OOD detection with continual learning to address challenges like catastrophic forgetting (CF) and data scarcity when new objects appear sporadically. OpenLD leverages linear discriminant analysis (LDA) and a pre-trained model to enable efficient OOD detection and incremental learning without CF.

**Strengths:**

1. Good sets of experiments across various backbones
2. Paper is easy to understand and flows well

**Weaknesses:**

1. Sounds like continual learning with pre-trained models and fine-tuning continually - which is the SoTA when it comes to transformer based CL methods.
2. There are existing generalized continual learning [1,2] frameworks that the authors overlook.
3. Authors complain most continual learning methods are replay based which is incorrect and must not be emphasized.
4. Small datasets are used for experiments, not representative of "open-world" as the authors emphasize often. Must use large datasets such as iNaturalist
4. Why aren't existing methods compared in the proposed setup? This is a major weakness.



[1] Generalized Class Incremental Learning - Fei Mi; Lingjing Kong; Tao Lin; Kaicheng Yu; Boi Faltings
[2] Online Class-Incremental Learning For Real-World Food Image Classification - Siddeshwar Raghavan, Jiangpeng He, Fengqing Zhu

**Questions:**

Please refer to the weaknesses.

---

### Note · Authors · 2024-11-15

I have read and agree with the venue's withdrawal policy on behalf of myself and my co-authors.